# Unusual Differences in the Pulmonary Histopathology of Mice after Intranasal Infection with Mycelial Propagules of *Histoplasma capsulatum* Strains Classified as LAm A2 and NAm 2 Phylogenetic Species

**DOI:** 10.3390/jof9100974

**Published:** 2023-09-27

**Authors:** Evelyn Pulido-Camarillo, Jorge H. Sahaza, Nayla de Souza Pitangui, Maria José S. Mendes-Giannini, Ana M. Fusco-Almeida, Armando Pérez-Torres, Maria Lucia Taylor

**Affiliations:** 1Departamento de Biología Celular y Tisular, Facultad de Medicina, Universidad Nacional Autónoma de México (UNAM), Av. Universidad 3000, Circuito Escolar s/n, Ciudad Universitaria, Ciudad de México 04510, Mexico; elyn_21@live.com.mx; 2Unidad de Micología, Departamento de Microbiología y Parasitología, Facultad de Medicina, Universidad Nacional Autónoma de México (UNAM), Ciudad de México 04510, Mexico; jhsahaza@hotmail.com; 3Departamento de Biología Celular e Molecular, Faculdade de Medicina, Universidade de São Paulo, Ribeirão Preto 14049-900, São Paulo, Brazil; napitangui@hotmail.com; 4Departamento de Análises Clínicas, Faculdade de Ciências Farmacêuticas, Universidade Estadual Paulista (UNESP), Araraquara 14800-903, São Paulo, Brazil; gianninimj@gmail.com (M.J.S.M.-G.); ana.marisa@unesp.br (A.M.F.-A.)

**Keywords:** *Histoplasma* genotypes, pulmonary response, pneumonia, granuloma, bronchiolar-associated lymphoid tissue

## Abstract

The ascomycete *Histoplasma capsulatum* is the causative agent of systemic respiratory mycosis histoplasmosis, which sometimes develops acute disseminated or chronic clinical forms, with the latter usually associated with granuloma formation. The present report shows differential histopathological changes in the pulmonary inflammatory response of mice infected intranasally with the mycelial morphotype of *H. capsulatum* strains with distinct genotypes, EH-46 and G-217B, classified as LAm A2 and NAm 2 phylogenetic species, respectively. Infected male BALB/c mice were sacrificed at different postinfection times, and their serial lung sections were stained with periodic acid–Schiff and analyzed via microscopy. In mice infected with the LAm A2 strain, the results showed progressive changes in the inflammatory infiltrate of the lung parenchyma during the first hours and days postinfection as well as granulomas with macrophages containing intracellular yeast cells, which prevailed at 14 and 21 days postinfection. Bronchiolar-associated lymphoid tissue was induced in mice infected with both strains, primarily in mice infected with the NAm 2 strain. Several lung sections from mice infected with the LAm A2 strain showed PAS-positive yeast cells aggregated in a perinuclear crown-like arrangement in macrophages from 3 h to 21 days postinfection. These findings highlight differences in the host pulmonary inflammatory response associated with distinct *H. capsulatum* species.

## 1. Introduction

Histoplasmosis is the most widely distributed systemic respiratory mycosis in the American continent. It is acquired via inhalation of aerosolized infective mycelial morphotype (M-phase) propagules of the dimorphic fungus *Histoplasma capsulatum*, mainly found in environments containing bat and bird guano, which favor M-phase growth. Clinical manifestations of the disease vary, ranging from a limited and mild respiratory infection to a life-threatening acute disseminated form, sometimes resulting in a fatal outcome. Under poorly defined circumstances, the disease evolves into a chronic form that develops tissue reactions represented by a well-organized cytoarchitecture known as a granuloma. The outcome of an *H. capsulatum* infection depends on the immune status of a specific host, the infective propagules’ multiplicity, the virulence, and the phylogenetic species of the fungal strains [1,2,3].

To date, the role of the innate immune response to *H. capsulatum* infection has not been carefully explored [1,2]. In contrast, the activation of cellular immunity through the CD4+ Th1 lymphocytes of the adaptive arm of the immune response has been preferentially analyzed in several studies of this host–fungus interaction. This response is, undoubtedly, the main mechanism related to fungal clearance in host-infected tissues. Most of the knowledge generated about this issue has been associated with experimental assays using murine models inoculated with the yeast morphotype (Y-phase), which is the parasitic morphotype presenting the main multiple virulence factors of this fungus [4,5,6,7]. However, the use of the Y-phase to study the course of the immune response to a histoplasmosis infection can occasionally cause misinterpretations in the progression of the first step of the infection because it is not the natural infective morphotype of *H. capsulatum*.

A few studies have processed the infective M-phase to explore the immune responses triggered by the host to this particular fungal morphotype, in the clinical evolution of histoplasmosis. New information regarding different sites and the time spent for the in vivo *H. capsulatum* M-to-Y transition was reported by Suárez-Álvarez et al. [8]. They detected the expression of phase-specific genes for M- and Y-phases using RT-PCR assays in tissue samples taken from the upper respiratory tract of infected mice. Thus, by utilizing this innovative genetic approach and by simulating a natural host infection, these authors found relevant data, which undoubtedly demonstrated that infective *Histoplasma* M-phase propagules spent 2–3 h to convert in the parasitic Y-phase. Furthermore, they also demonstrated that this fungus can also utilize other extrapulmonary sites to initiate its M-to-Y transition, where the link between innate and adaptive immune responses occurs, such as the mucosal-associated lymphoid tissues in the upper respiratory tract [8].

Sahaza et al. [2] also used the infective M-phase to trigger the immune response in mice, demonstrating that the highest levels of several innate and adaptive proinflammatory cytokines, detected in pulmonary homogenates of adult male BALB/c mice infected with the fungal M-phase, were primarily associated with a particular *H. capsulatum* strain isolated in Mexico (EH-46), which was first classified as a Latin American LAm A phylogenetic species, according to Kasuga et al. [9], and later renamed LAm A2 by Teixeira et al. [10]. As stated by Sahaza et al. [2], these cytokine levels contrasted with those obtained using the G-217B *H. capsulatum* strain isolated in the United States of America (USA), which belongs to the NAm 2 phylogenetic species and can also be cataloged as *Histoplasma ohiense* sp. nov., according to Kasuga et al. [9] and Sepúlveda et al. [11], respectively. Thus, the morphotype and the phylogenetic species of *H. capsulatum* are some of the major attributes that certainly participate in histoplasmosis manifestations and in the fate of this type of host–parasite interaction.

Durkin et al. [12] documented the role of different fungal genotypes regarding the dissimilarities in the outcome of the disease and in the histopathological damage, which were triggered by *H. capsulatum* strains from Latin America (earlier classified as classes 5 and 6) in mice infected intratracheally with the Y-phase, in contrast to those infected with a strain from North America (class 2). Concerning the virulence of distinct phylogenetic strains isolated in North America, Sepúlveda et al. [3] reported differences in the fungal burden, disease progression, and cytokine responses in mice infected intranasally with yeast cells of *H. capsulatum*.

In human histoplasmosis, Karimi et al. [13] reported clinical differences between AIDS-associated histoplasmosis patients from the USA and those from Brazil, highlighting the most frequent skin lesions in Brazilian patients infected with classes 5 and 6 genotypes of *H. capsulatum* strains, in contrast to patients from the USA infected with the class 2 *H. capsulatum* genotype. Mucocutaneous manifestations of histoplasmosis are frequently observed in Brazilian patients and are caused by specific strains with unique pathogenic characteristics within the phylogenetic species of *H. capsulatum* from Latin America, which might explain its increased dermatotropism [14]. Although in human patients, the host genetic characteristics could participate in the fate of the interaction with the fungus performance, in experimental models, these characteristics can usually be homogenized; also, the genetic attributes of the fungus involved in the infective process of the host should be considered as a relevant factor.

Currently, several authors emphasize the fact that *H. capsulatum* realizes its dimorphic transition when it reaches alveoli, within their macrophages. However, our research group has found that the upper respiratory tract of the infected host, including the nasal-associated lymphoid tissue, plays an important role in the in vivo *H. capsulatum* M-to-Y transition and in the immune response to this pathogen, taking part in the initial mechanisms activated by the defense of the hosts against airborne infective M-phase propagules [8]. During the early stages of the innate immunity (probably up to 5 days postinfection), no real evidence of chronic inflammation characterized by the development of granulomatous reactions could be found [2,8]. These reactions are formed in histoplasmosis after the fungal dimorphic transition to the Y-phase has occurred in respiratory microenvironments of the host, once the adaptive chronic inflammatory reaction is established. The development of a granuloma throughout the histoplasmosis infectious process was monitored in an elegant study conducted by Heninger et al. [15]. These authors evaluated granuloma formation at different times after intraperitoneal and intranasal infection of mice, using the Y-phase of the G-217B *H. capsulatum* strain (NAm 2 phylogenetic species or *H. ohiense*); after the seventh day postinfection, histopathological findings revealed the presence of granulomas containing yeast cells in the lungs and liver of the infected mice, irrespective of the infection route.

Considering all the abovementioned antecedents and the fact that *H. capsulatum* affects the lungs as a main target organ, the present study aimed to record the most exciting features in the lung’s cytoarchitecture during the inflammatory response associated with differential histopathological changes generated by the intranasal infection of mice with the M-phase propagules of two different phylogenetic strains of this fungal pathogen.

## 2. Materials and Methods

### 2.1. Histoplasma Strains

Two strains from different phylogenetic species were selected for this study to develop experimental murine histoplasmosis. The EH-46 strain, which was isolated from a patient infected in a histoplasmosis outbreak in the state of Guerrero, Mexico, was deposited in the *H. capsulatum* Culture Collection of the Fungal Immunology Laboratory (www.facmed.unam.mx/histoplas-mex/) at the Departamento de Microbiología y Parasitología, Facultad de Medicina, Universidad Nacional Autónoma de México (UNAM). This collection is registered in the database of the World Data Centre for Microorganisms under number LIH-UNAM WDCM817 (http://www.wfcc.info/ccinfo/index.php/strain/display/817/fungi/). The G-217B strain, which was isolated from a patient from Louisiana (USA), was deposited in the American Type Culture Collection (ATCC-26032) and is usually considered a reference for *H. capsulatum*. The EH-46 and G-217B strains are classified as belonging to the LAm A2 (isolated in Latin America) and NAm 2 (isolated in North America) phylogenetic species, as reported by Teixeira et al. [10] and Kasuga et al. [9], respectively.

### 2.2. Production of Infective M-Phase Propagules

Mycelium cultures of each *H. capsulatum* strain at 28 °C were processed in potato dextrose agar (Bioxón, Becton Dickinson, Mexico City, CDMX, Mexico) to produce abundant M-phase propagules. Each culture was processed to obtain mainly microconidia and small hyphal fragments as described by Sahaza et al. [2]. Briefly, the M-phase of each strain was harvested in saline solution (SS) and centrifuged at 800× *g*, for 10 min at 4 °C, to remove macroconidia and large aggregates of mycelium. The inoculum of each strain was suspended in SS and standardized as homogeneously as possible, and it contained approximately 90% microconidia and 10% hyphal fragments. Each inoculum was counted in a hemocytometer via optical microscopy and adjusted to 3 × 10^6^ infective propagules/40 µL. Inoculum viability was determined using the trypan blue (0.5%) exclusion test, and the infective inoculum was confirmed via culturing on non-supplemented BHI agar (Bioxón) at 28 °C, using the determination of colony-forming units.

### 2.3. Mice Infection

Six-week-old male BALB/c inbred mice were maintained under a controlled temperature (23 to 24 °C) with a 12 h light/dark cycle and received balanced feed and water ad libitum. Mice were infected intranasally with a fungal inoculum in SS suspension adjusted to 3 × 10^6^/40 μL of M-phase propagules. Mice were anesthetized by injecting a mixture of ketamine (100 mg/kg)/xylazine (10 mg/kg) (Mexican Official Guide, NOM-062-ZOO1999) intramuscularly to facilitate intranasal infection, dispensing 10 μL of the inoculum into each animal nostril at intervals of approximately 30 s, until completion of 40 μL. For each *H. capsulatum* strain, 24 mice were infected, and 3 animals were sacrificed per infection time point: 1, 3, 24, and 48 h, as well as 5, 7, 14, and 21 days after infection. As noninfected controls, three animals per infection time received 40 μL of sterile SS. Infected and control mice were observed daily for murine histoplasmosis signs, such as hirsute hair, cachexia, weight loss, and immobility. The euthanasia of animals was conducted according to the guidelines of the American Veterinary Medical Association, using 5% isoflurane followed by cervical dislocation.

### 2.4. Ethic Statements

All procedures involving the care and use of animals were performed in accordance with the recommendations of the Animal Care and Use Committee of the Faculty of Medicine (UNAM) in compliance with the project number 166052 from CONACyT-Mexico, following the recommendations of the Mexican Official Guide (NOM062-ZOO-1999) in agreement with the National Research Council (US) Committee (Guide for the Care and Use of Laboratory Animals from the National Institute of Health of the USA, available from https://www.ncbi.nlm.nih.gov/books/NBK54050.

### 2.5. Histological Processing

All mice were sacrificed using isoflurane, in compliance with the Mexican Official Guide (NOM-062-ZOO1999). To prevent alveolar collapse, the lungs were fixed via intratracheal perfusion with 500 µL of 4% paraformaldehyde diluted in SS. After thoracotomy, the lungs were dissected and then submerged in the same fixative solution for a minimum period of 48 h. Then, the lungs were rinsed in tap water, dehydrated through ascending grades of ethanol, cleared in xylene, and embedded in paraffin with a particular orientation to obtain complete sections (4 μm thick) from the same plane of the inferior lobes of both lungs. Serial or consecutive tissue sections were stained with periodic acid–Schiff (PAS) and analyzed using a BX50 Olympus microscope equipped with a digital camera and Infinity Analyze software, v6.3.0.

## 3. Results

### 3.1. Pulmonary Findings in Mice from 1 to 48 h Postinfection with Histoplasma Strains

First, at 1 h postinfection, histopathological findings revealed tissue changes in the lungs of BALB/c mice infected with the M-phase of the EH-46 and G-217B *H. capsulatum* strains, showing evident increased cellularity in the alveolar walls. PAS-positive fungal cells, compatible with macroconidia of *H. capsulatum*, were observed in the alveolar lumen of mice infected with the EH-46 and G-217B strains (Figure 1A,B). Neutrophils could be observed in different microscopic fields, predominantly in the pulmonary parenchyma, concomitant with an important thickness of interalveolar septa, primarily in mice infected with the G-217B strain (Figure 1B). Intracellular PAS-positive fungal cells, compatible with yeast- or microconidia-like cells, were observed within an alveolar macrophage from a mouse infected with the EH-46 strain (Figure 1C); in addition, eosinophilic materials (protein deposits) were found within some alveolar spaces.

At 3 h postinfection, the alveolar spaces were well conserved, suggesting a good ventilation process. However, the inflammatory infiltrate of the pulmonary parenchyma was evident and was associated with the major thicknesses of the alveolar septa in the lungs of mice infected with both fungal strains (Figure 2A,B), although this finding was more marked in mice infected with the G-217B strain (Figure 2B). At this postinfection time, based on the study published by Suárez-Álvarez et al. [8], only yeast cells were observed within alveolar macrophages with both fungal strains, mainly in mice infected with the EH-46 strain (Figure 2C). Also, at this postinfection time, the aggregation of intracellular yeast cells in crown-like arrangements surrounding the macrophage nucleus (Figure 2C,D) was notable. In addition, PAS-positive fungal cells compatible with the macroconidia of *H. capsulatum* were revealed again in the lung sections of mice infected with the EH-46 and G-217B strains (Figure 2E,F).

In the sequence of histological events, at 24 h postinfection, the inflammatory infiltrate of the alveolar spaces was more marked, and it contained numerous neutrophils and a few alveolar macrophages, which were mainly observed in mice infected with the EH-46 strain (Figure 3A). In contrast, in mice infected with the G-217B strain, the inflammatory infiltrate diminished considerably, and in most microscopic fields, the histological architecture was similar to that found in normal pulmonary parenchyma (Figure 3B). Intracellular yeast cells were observed either in neutrophils or in alveolar macrophages from mice infected with both *H. capsulatum* strains, predominantly with the EH-46 strain (Figure 3C). The absence of neutrophils in the alveolar wall and the presence of scarce macrophages with intracellular yeast cells were primarily associated with mice infected with the G-217B strain (Figure 3D).

After 48 h postinfection, the inflammatory infiltrate persisted in the lungs of the mice infected with both strains (Figure 4A,B), and the cumulus of neutrophils and mononuclear cells were observed around the bronchiolar wall and blood vessels, which is compatible with the initial step of bronchiolar-associated lymphoid tissue (BALT) development (Figure 4A). PAS-positive intracellular yeast cells were observed in macrophages and neutrophils from mice infected with both fungal strains, highlighting yeast cells aggregated in crown-like perinuclear arrangements in infected macrophages when the EH-46 strain was used (Figure 4C,D). At this postinfection time, only a few intracellular yeast cells within macrophages and in the cumulus of neutrophils were observed in lung sections from a mouse infected with the G-217B strain.

### 3.2. Pulmonary Findings in Mice from 5 to 21 Days Postinfection with Histoplasma Strains

When the infection persisted for several days, the damage to the pulmonary parenchyma caused by each of the two fungal strains used could be gradually distinguished. At 5 and 7 days postinfection, the alveolar inflammatory infiltrate remained persistent, particularly in lung samples from mice infected with the EH-46 strain (Figure 5A,B), where neutrophils with intracellular yeast cells predominated. In contrast, a minor inflammatory infiltrate and the occurrence of detachments of the bronchiolar epithelium were observed only in mice infected with the G-217B strain (Figure 5C,D). PAS-positive yeast cells were revealed in the inflammatory infiltrate caused either by the EH-46 or the G-217B strains, as noted in the detached bronchiolar epithelium produced by the G-217B strain infection (Figure 5D).

At 14 days postinfection, histopathological dissimilarities were evident in many pulmonary fields concerning the inflammatory response in mice infected with one or the other genetically different strains of *H. capsulatum*. In a mouse infected with the EH-46 strain, the cumulus of mononuclear cells (macrophages and lymphoid cells) could be identified in a large inflammatory infiltrate, swelling the alveolar wall, collapsing the alveolar lumen, and surrounding the bronchioles and branches of pulmonary arteries, as clearly shown in a sequential microscopic field in Figure 6A,B. Clear, well-organized granulomas were also observed in these infected mice. Particularly, a granuloma was located next to the bronchiole wall, affecting the adjacent alveoli in the lung sample of a mouse infected with this strain (Figure 6A,B). In addition, neutrophils and several macrophages with intracellular yeast cells can be observed in different microscopic fields of lung samples from mice infected with the EH-46 strain (Figure 6C–F), and again, the notable presence of well-formed PAS-positive yeast cells aggregated in a crown-like intracellular arrangement was evident in an infected macrophage (Figure 6F). Importantly, most findings described with the EH-46 strain are compatible with the process of granuloma formation (Figure 6A–E). In contrast, in mice infected with the G-217B strain, the inflammatory infiltrate was minor, without granuloma in development, with scarce neutrophils and macrophages, and with the presence of BALT without evidence of fungal cells, as shown in a particular sequential microscopic field (Figure 6G,H). Macrophages with a crown-like perinuclear arrangement of *H. capsulatum* yeast cells were scarcely observed with this strain. Overall, microscopic fields of other mice lung sections infected with the G-217B strain always showed fewer intracellular yeast cells than those found in lung sections from mice infected with the EH-46 strain.

At 21 days postinfection, the histological inflammatory reactions related to both *H. capsulatum* strains were very different. The lungs of mice infected with the EH-46 strain showed an important presence of mature granulomas in their sections, which were represented by dense mononuclear inflammatory infiltrates and scarce neutrophils surrounding the bronchioles and blood vessels, as shown in singular sequential microscopic fields of Figure 7A,B. Numerous macrophages containing intracellular yeast cells were observed at this postinfection time in lung sections when the EH-46 strain was used to infect mice. However, in mice infected with the G-217B strain, scarce intracellular yeast cells predominated, and the inflammatory infiltrate remained without substantial changes, although it was more evident in the alveolar wall and in the BALT (Figure 7C,D); observations of several lung sections of mice infected with this strain always indicated the absence of granulomas.

Once more, alveolar macrophages with perinuclear yeast cells in a crown-like arrangement (Figure 7E) were observed in the lung sections of mice infected with the EH-46 strain and in the lung section of a mouse infected with the G-217B strain (Figure 7F).

## 4. Discussion

The host inhalation route certainly participates in the initial recognition of the infective M-phase of *H. capsulatum* by promoting the activation of multifaceted mechanisms triggered by the host–fungus interplay, which include the immune response of the host respiratory system and fungal strategies. Regarding the *H. capsulatum* strategies, the fast in vivo morphotype transition to the parasitic-virulent Y-phase is necessary for the establishment of the infection and subsequent dissemination, via lymph drainage and lymph nodes. Notably, we have demonstrated that fungal dissemination could occur before it affects the lung as the main target organ [8].

To investigate the inflammatory reaction generated by distinct genotypes from two *H. capsulatum* strains belonging to different phylogenetic species, LAm A2/EH-46 (Latin America) and NAm 2/G-217B (North America), we used an intranasal inoculum of fungal M-phase propagules in a murine model to simulate a natural infection. In this study, intracellular yeast cells were primarily observed after 3 h postinfection in macrophages of mice infected with the EH-46 strain, particularly in the bronchiole-alveolar lumen. However, macrophages with intracellular yeast cells were less frequent in mice infected with the G-217B strain, which was mainly found in the BALT.

Studies associated with human clinical cases and animal models have found differences in virulence and gene expression, as well as in the pathogenesis of the disease, among *H. capsulatum* strains from different genotypes and phylogenetic species [1,2,3,12,13,14,16].

Several steps of the host response are activated during the host–fungus interaction, involving various levels of complexity, where the inflammatory process is critical for the host defense; however, sometimes, this inflammatory response progresses into chronic inflammation. In this setting, during the progression of the lung chronic inflammatory reaction, the associated immunopathology is characterized by changes in tissue architecture and variable cellularity in pulmonary lesions with well-organized granulomas, with or without necrosis development [12,15]. In the present paper, important histopathological differences were described in the lung inflammatory response of mice infected intranasally with M-phase propagules of *H. capsulatum* strains from distinct phylogenetic species, regarding the presence of pneumonia, the development of granulomas, and the BALT induction. The latter is an unexpected finding because BALT is not constitutive in mice [17]. According to our histopathological observations, after 14 and 21 days postinfection, granuloma development was clearly associated with the EH-46 strain, whereas BALT induction was associated with a lesser diffuse inflammatory response related to the G-217B strain infection. These findings support the existence of relevant differences in the pulmonary histopathology produced by each of the two *H. capsulatum* strains used in the present study, which could be crucial for the formation of granulomas in mice infected with the EH-46 strain. Overall, granuloma plays a protective role in the infected host by avoiding the progression of the fungal infection; however, depending on the course of some chronic infections, the granuloma could subvert the host defense, involving several organs and leading to tissue damage with fibrotic processes. Moreover, granulomas produced by *H. capsulatum* infection develop hypoxic microenvironments, which favor the infectious process [18]. To change this hypoxic microenvironment, host macrophages activate a transcription factor, the hypoxia-inducible factor (HIF)-1a, which abrogates the progression of *H. capsulatum* infection by controlling IL-10 production and, consequently, circumvents the negative effect of this cytokine on the antimicrobial activity of macrophages [19].

The presence of alveolar proteinosis (eosinophilic protein deposits) in the alveoli lumen was observed in a mouse infected with the EH-46 strain. Curiously, in human clinical cases, pulmonary alveolar proteinosis has also been documented in acute disseminated histoplasmosis [20].

Initially, progressive changes in the pulmonary parenchyma with increased cellularity (macrophages and neutrophils) compatible with pneumonitis were observed in mice infected with both strains at 1 h postinfection. According to our observations, *H. capsulatum* intracellular PAS-positive fungal cells, compatible with yeast- or microconidia-like cells, were first observed within alveolar macrophages at 1 h postinfection, whereas at 3 h postinfection, we recognized that all fungal cells were converted to yeast cells, as this time matched with the time required by this fungus to complete its dimorphic transition in vivo [8]. PAS-positive structures like macroconidia of *H. capsulatum* were only observed until 3 h postinfection. Therefore, from this time forward, all the fungal structures observed in lung histological sections were assumed as yeast cells, in accordance with the findings reported in a similar mice model by Suárez-Álvarez et al. [8].

As stated in our results, during the first 3 h postinfection with the EH-46 strain, the scarce number of macrophages with intracellular yeast cells contrasts with the number found for the G-217B strain; however, after 24 h postinfection, this relation changed for each *H. capsulatum* strain.

At 5–7 days, mainly in mice infected with the EH-46 strain, persistent inflammatory infiltrates were observed in their lung samples, which could evolve toward granulomas in development. Well-structured lung granulomas containing several yeast cells, predominate at 14 and 21 days postinfection, in mice infected with the EH-46 *H. capsulatum* strain. However, in mice infected with the G-217B strain, induced BALT prevailed.

Durkin et al. [12] showed that strains originating from Latin America (classes 5 and 6) caused more tissue damage and death in B6C3F mice infected intratracheally with sublethal doses of *H. capsulatum* yeast cells than in mice infected with a strain (class 2) from North America. They also stated that yeast cells were more abundant in B6C3F mice infected with *H. capsulatum* strains from Latin America. In addition, at 14 days postinfection, they showed the presence of granulomas and early caseous necrosis in mice infected with these strains, whereas mice infected with strain class 2 developed only lung inflammatory processes without evidence of granulomas or necrosis. No multinucleated giant cells were reported by these authors in the lung histopathology from mice infected with both strains. Our results using a LAm A2 *H. capsulatum* strain matched well with data reported by Durkin et al. [12], supporting the presence of important changes in the host response due to the genetic characteristics of *H. capsulatum* strains.

Regarding the time required for granuloma formation in an experimental histoplasmosis model, our findings are also consistent with those reported previously by Heninger et al. [15], who suggested that during the development of granuloma at different postinfection times in C57BL/6 mice, the appearance of well-structured granulomas in the lung and liver begin at day 7 postinfection and persist through days 10 and 14 in both organs. Similar times were observed by our team for granuloma development when we used the EH-46 *H. capsulatum* strain.

A previous report by Sahaza et al. [2], highlighted differences in granuloma formation based on the description of an original panoramic or a semi-microscopic lung section from mice infected with the LAm A2/EH-46 and NAm 2/G-217B strains; granulomas were well developed at days 14 and 21 postinfection, decreasing at day 28 after infection with the EH-46 strain. In contrast, mice infected with the G-217B strain did not show any images suggestive of granulomas. In our present study, we performed the same experimental infection conditions described by Sahaza et al. [2]; however, unlike us, they did not conduct histopathological observations to discriminate between lung tissue injuries generated by the genetically distinct *H. capsulatum* strains used.

Other authors have contributed to the description of the histoplasmosis course depending on the *Histoplasma* species, such as Sepúlveda et al. [3], who compared the disease progression using one *Histoplasma* strain from the Panama lineage and two *Histoplasma* strains representative of the NAm 1 and NAm 2 phylogenetic species to infect mice intranasally with low or high sublethal doses of *H. capsulatum* yeast cells. They showed changes in clinical manifestations (weight loss), pathogenesis (lung inflammatory infiltrate), host response (cytokine production), and disease resolution, depending on the inoculum size, the virulence, and the *H. capsulatum* phylogenetic group. Their results highlight that the lineage from Panama showed the highest virulence in mice exposed to a lower yeast cells inoculum, in contrast to the strains from the NAm 1 and NAm 2 groups. However, using the same experimental conditions, Jones et al. [16] reported that the *Histoplasma* strain from the NAm 2 phylogenetic group is mostly found within alveolar macrophages and produces progressive lung inflammation in infected mice.

Some findings observed during the present study should be discussed, such as the presence of intracellular *H. capsulatum* yeast aggregates surrounding the infected macrophage nucleus, which started hours after the infection and persisted for several days postinfection. This yeast aggregate construction is compatible with previous observations reported by Pitangui et al. [21] in cultured murine alveolar macrophages (AMJ2-C11 cell line) infected with *H. capsulatum* yeast cells. These authors were the first to describe this intracellular conformational architecture of yeast cells in aggregation, and they proposed that this phenomenon may be related to injuries in the nuclei of the infected host cells, causing their DNA fragmentation and apoptosis. According to our results, this repetitive perinuclear arrangement of the *H. capsulatum* Y-phase was primarily observed in macrophages of mice infected with the EH-46 strain. These yeast cell aggregates within lung macrophages emerged in mice inoculated with fungal M-phase propagules, and their presence was conspicuous after 3 h postinfection, corresponding to the time required to complete the M-to-Y transition under a simulated natural infection [8]. Thus, this is the first report describing the *H. capsulatum* Y-phase (parasitic phase) forming intracellular perinuclear organized yeast cell structures in macrophages of the pulmonary parenchyma of mice infected intranasally with the *H. capsulatum* M-phase (infective phase).

In the human clinical course of histoplasmosis, the impact of fungal genetic diversity associated with *H. capsulatum* strains from different phylogenetic species is still an emerging field to be explored in the study of this fungal disease. Reports by Damasceno et al. [22,23] referred to an unusual human coinfection with different *H. capsulatum* genetic groups (Northeast BR1 and Northeast BR2), and they also showed differences in their mating types (MAT1-1 and MAT1-2) in the same AIDS–histoplasmosis patients. At this moment, no experimental information has explained the consequence of this type of coinfection; however, such coinfection can reasonably be speculated to interfere with the optimal host immune response and to trigger a more aggressive pathogenesis as well as a threatening outcome.

## 5. Conclusions

The present results may contribute new information to a better understanding of *H. capsulatum*-induced lung inflammatory reactions associated with pneumonia, granuloma, or BALT induction. They also provide a baseline for researchers conducting further studies to comprehend the lung inflammatory response, specifically, how granulomas are formed in infected hosts that can discriminate infective mycelial propagules from genetically diversified *Histoplasma* species. In addition, our findings suggest that intranasal infection of mice using mycelial propagules of *H. capsulatum* better reproduces the natural course of fungal infection and lung histopathology with granuloma and BALT developments in histoplasmosis.

## Figures and Tables

**Figure 1 jof-09-00974-f001:**
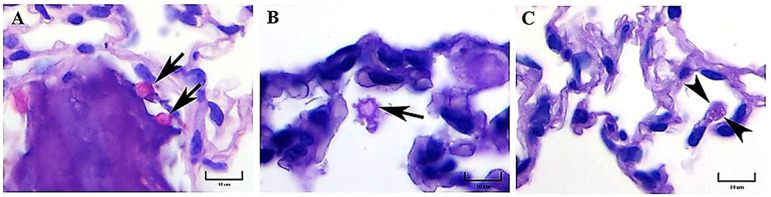
Representative histopathology of lung sections of BALB/c mice at 1 h postinfection. PAS-positive macroconidium-like fungal structures of *H. capsulatum* measuring 6–8 μm in diameter were observed in the alveolar space of mice infected with either the EH-46 (**A**) or the G-217B (**B**) strains (arrows). PAS-positive fungal cells are shown intracellularly in a macrophage from a mouse infected with the EH-46 strain (**C**, arrowheads). PAS staining. Scale bars = 10 µm (**A**–**C**).

**Figure 2 jof-09-00974-f002:**
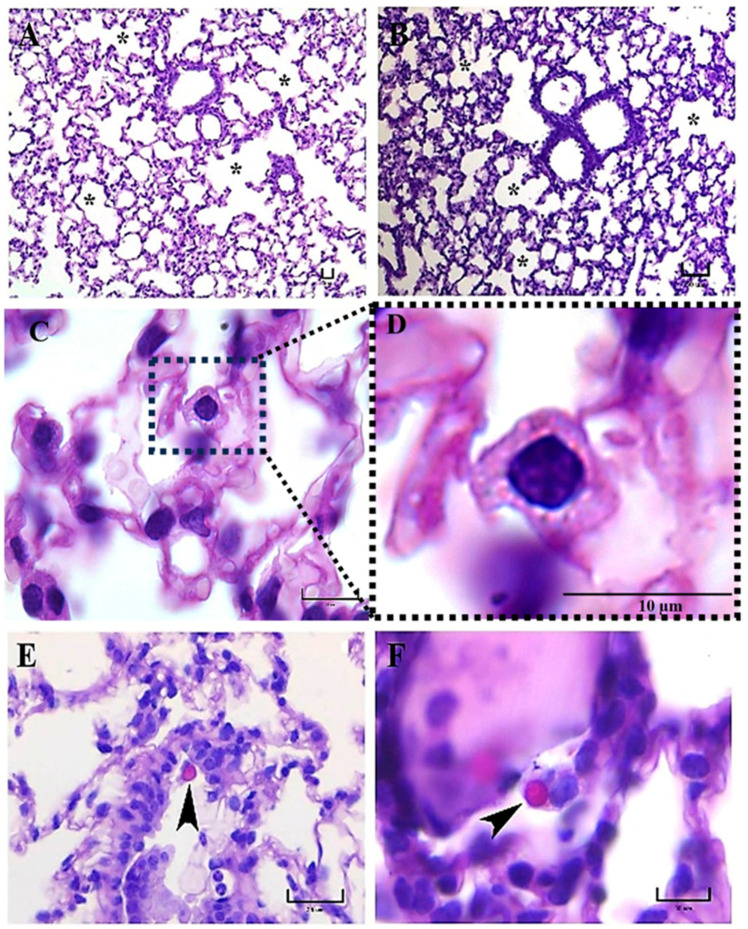
Representative histopathology of lung sections of BALB/c mice at 3 h postinfection. Well-aerated alveolar spaces (asterisks) were observed in mice infected with either the EH-46 (**A**) or the G-217B (**B**) *H. capsulatum* strains, although inflammatory infiltrates of the pulmonary parenchyma (at alveolar walls) were also observed with both strains. In some lung sections of mice infected with the EH-46 strain, PAS-positive aggregates of intracellular yeast cells exhibited crown-like arrangements surrounding the alveolar macrophage nucleus (**C**,**D**). Macroconidium-like structures of *H. capsulatum* (arrowheads) were observed in the lung sections of mice infected with the EH-46 (**E**) and G-217B strains (**F**). PAS staining. Scale bars = 20 µm (**A**); 50 µm (**B**); 10 µm (**C**,**D**,**F**); and 25 µm (**E**).

**Figure 3 jof-09-00974-f003:**
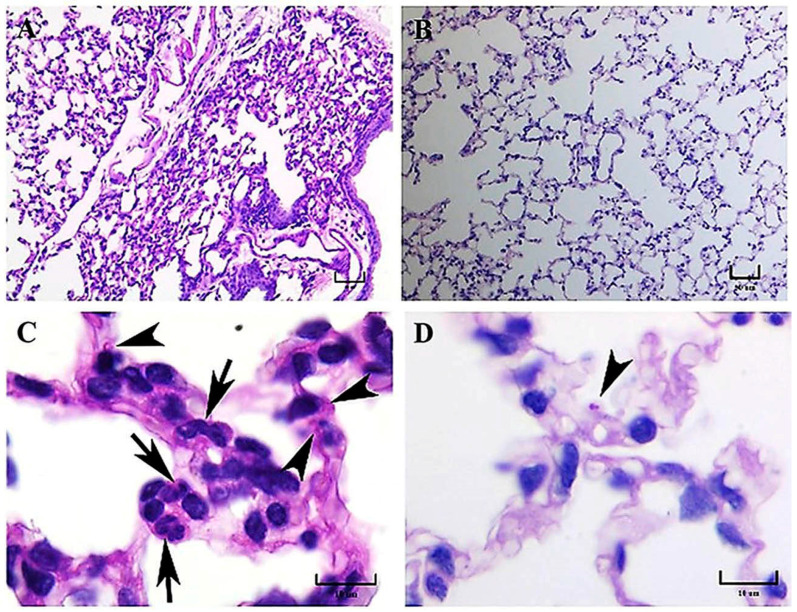
Representative histopathology of lung sections of BALB/c mice at 24 h postinfection. Inflammatory infiltrates containing numerous neutrophils and the presence of focal-reduced aerated spaces were observed in a mouse infected with the EH-46 *H. capsulatum* strain (**A**). Histological architecture similar to normal pulmonary parenchyma was found in a mouse infected with the G-217B strain (**B**). Intracellular yeast cells in macrophages (arrowheads) and neutrophils (arrows) were observed in a mouse infected with the EH-46 strain (**C**). In the alveolar wall, neutrophils were absent, and macrophages with intracellular yeast cells were scarce (arrowhead) in mice infected with the G-217B strain (**D**). PAS staining. Scale bars = 50 µm (**A**,**B**); 10 µm (**C**,**D**).

**Figure 4 jof-09-00974-f004:**
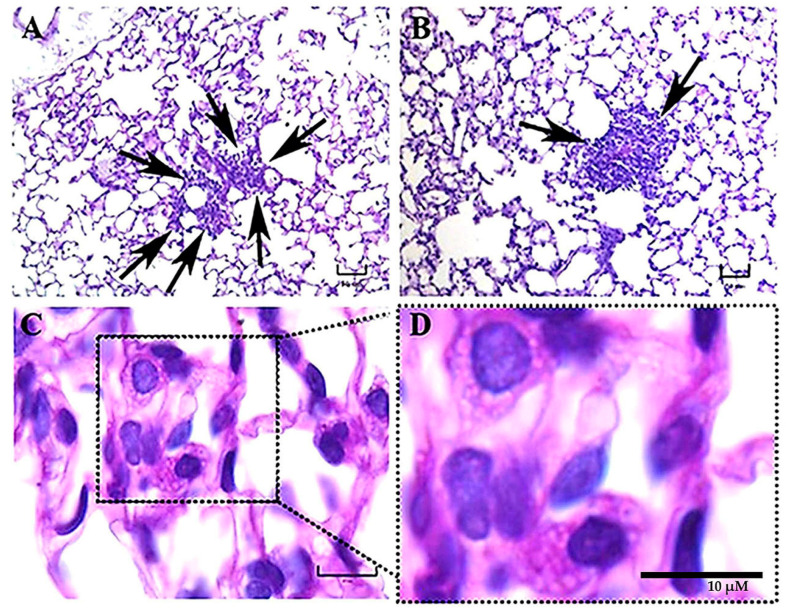
Representative histopathology of lung sections of BALB/c mice at 48 h postinfection. A bronchiole-alveolar wall near blood vessels infiltrated by numerous neutrophils and mononuclear cells, resembling BALT in development (arrows), was observed in a mouse infected with the EH-46 *H. capsulatum* strain (**A**). Dense inflammatory reactions (arrows) were observed in the alveolar wall of a mouse infected with the G-217B strain (**B**). Intracellular yeast cells in lung sections from a mouse infected with the EH-46 strain showed aggregations in a crown-like arrangement surrounding the macrophage nucleus (**C**,**D**). PAS staining. Scale bars = 50 µm (**A**,**B**); 10 µm (**C**,**D**).

**Figure 5 jof-09-00974-f005:**
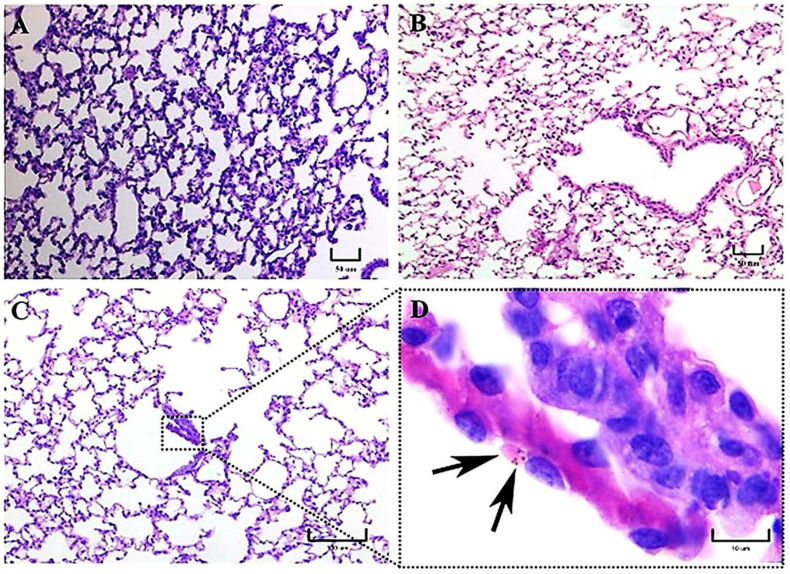
Representative histopathology of lung sections of BALB/c mice at 5 and 7 days postinfection. Persistent inflammatory infiltration on the alveolar walls was evident at 5 days (**A**) and 7 days (**B**) postinfection with the EH-46 *H. capsulatum* strain. In the lung sections, at 5 days postinfection with the G-217B strain, the inflammatory infiltrate was sporadic, and the microscopic field showed a detached bronchiolar epithelium (**C**,**D**), exhibiting few intracellular yeast cells (**D**, arrows). PAS staining. Scale bars = 50 µm (**A**,**B**); 100 µm (**C**); 10 µm (**D**).

**Figure 6 jof-09-00974-f006:**
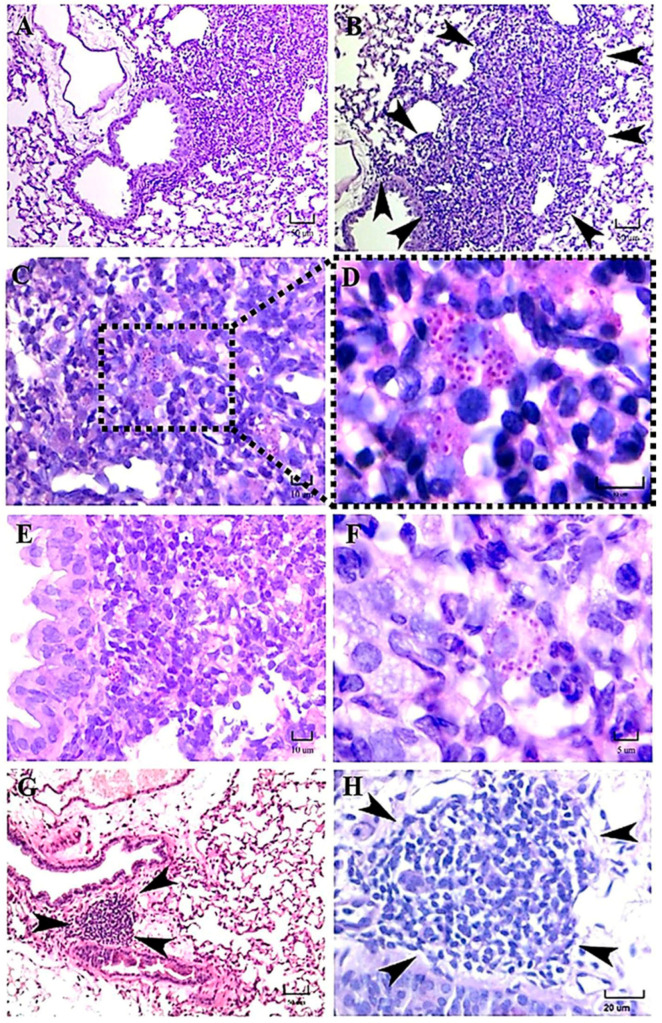
Representative histopathology of lung sections of BALB/c mice at 14 days postinfection. Regarding mice infected with the EH-46 strain, a large lung granuloma was observed close to the bronchiole wall, affecting the neighboring alveoli (**A**,**B**). In addition, several intracellular yeast cells were found in different microscopic fields of lung samples from a mouse (**C**–**F**); in particular, a crown-like arrangement of perinuclear yeast cells was observed in a macrophage from a mouse infected with the EH-46 strain (**F**). BALT hyperplasia, without evidence of yeast cells, was identified in a mouse infected with the G-217B strain (**G**,**H**, arrowheads). PAS staining. Scale bars = 50 µm (**A**,**B**,**G**); 10 µm (**C**–**E**); 5 µm (**F**); and 20 µm (**H**).

**Figure 7 jof-09-00974-f007:**
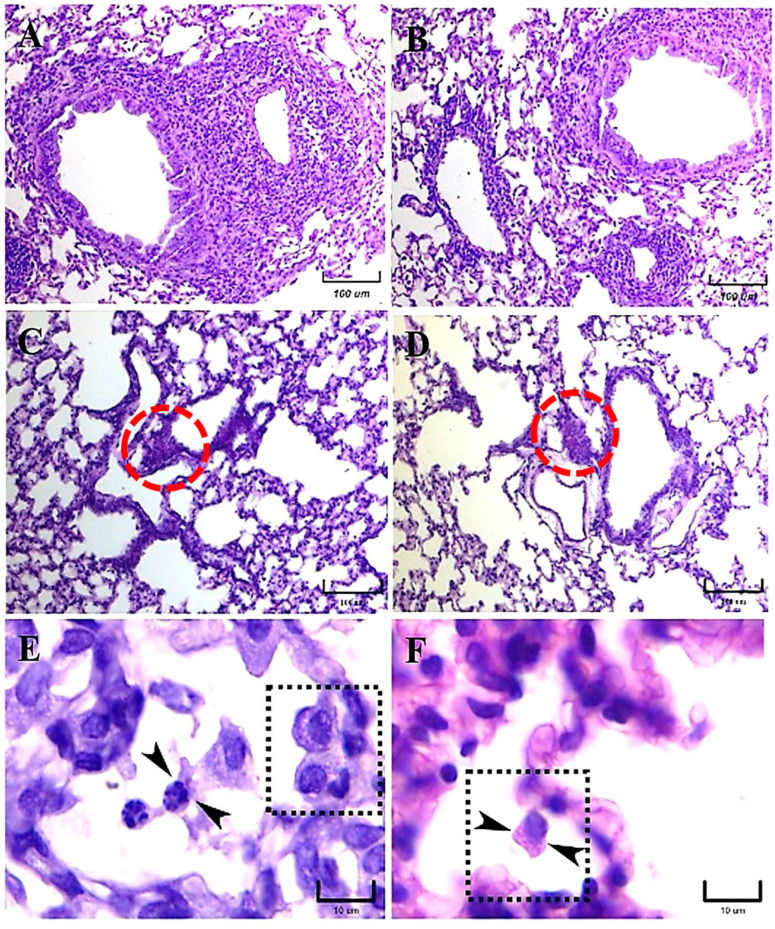
Representative histopathology of lung sections of BALB/c mice at 21 days postinfection. Well-developed peribronchiolar and perivascular granulomas were observed in mice infected with the *H. capsulatum* EH-46 strain (**A**,**B**). In contrast, lung sections of mice infected with the G-217B strain were devoid of granulomas; however, alveolar wall inflammatory infiltrates were clearly shown, and the presence of BALT was identified (**C**,**D**, red dashed circles). Intracellular yeast cells in neutrophils (**E**, arrowheads) and a well-formed crown-like arrangement of perinuclear yeast cells in an alveolar macrophage (**E**, dashed square) were observed in mice infected with the EH-46 strain. In addition, an alveolar macrophage with yeast cells in a crown-like arrangement was also observed in a lung section of a mouse infected with the G-217B strain (**F**, arrowheads in dashed square). PAS staining. Scale bars = 100 µm (**A**–**D**); 10 µm (**E**,**F**).

## Data Availability

The corresponding authors (M.L.T. and A.P.-T.) declare that all research data contained in this article are available to all potential readers and scholars by requesting these data appropriately through their corresponding emails.

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
