# Peer review of "Unusual Differences in the Pulmonary Histopathology of Mice after Intranasal Infection with Mycelial Propagules of Histoplasma capsulatum Strains Classified as LAm A2 and NAm 2 Phylogenetic Species"

_jof, 2023, doi:10.3390/jof9100974_

Round 1
Reviewer 1 Report
Titles
The present article “Unusual Differences in the Pulmonary Histopathology of Mice after Intranasal Infection with Mycelial Propagules of Histoplasma capsulatum Phylogenetic Species from Latin America (LAm A2) and North America (NAm 2)”, had many problems beginning from the Titles.
Please change the title to:
Differences in the Pulmonary Histopathology of Mice after Intranasal Infection with Mycelial Propagules of Histoplasma spp. from Histoplasma suramericanum (LAm A2) and Histoplasma ohiense (NAm 2)
Another of my worries with the present work if the type of inoculum used, mycelial propagules are unreliable, because not all the mycelial fragments (depending of its size) are able to reach the alveoli and produce infection, that is the reason that most researchers use mycelial microconidia, or yeast. Although in the material and methods it specifies that the inoculum is mainly microconidia, it should indicate what is the percentage, 60% - 90% ?
Regarding the part of “Histoplasma capsulatum Phylogenetic Species” it doesn’t make any sense. Should be: “Histoplasma spp.” The two isolates used in the present work have been classified as H. ohiense (G-217B, Nam 2) and H. suramericanum (EH-46, Lam A2). The reference 11(Sepulveda VE, Marquez R, Turissini DA, Goldman WE, Matute DR. Genome Sequences Reveal Cryptic Speciation in the Human Pathogen Histoplasma capsulatum. mBio. 2017;8(6). 10.1128/mBio.01339-17), that use complete genomes of Histoplasma, validates the original classification of Kasuga et al. (Ref. 9), but give proper names to several of the clades original described by Kasuga et. al. and should be use, at least the authors of the present work have data supporting the actual classification presented in reference 11.
Abstract
The same changes should be done in the abstract, using Histoplasma spp. instead of Histoplasma capsulatum. Also use H. suramericanum(EH-46, Lam A2) and H. ohiense (G-217B, Nam 2) and drop the use of Phylogenetic species since these species have already been described. At the end of the abstract change Histoplasma genotypes for Histoplasma species.
Introduction
Similar changes should be introduced in this section, referring to Histoplasma spp. or Histoplasma strains. Drop the use of Phylogenetic species that don’t make any sense, as mentioned above.
In the initial part, where the description of the disease and its clinical manifestations are made, the reference that should be used, is one related with the subject, such a recent Medical Mycology book, or good book on infectious diseases, in its defect a review on Histoplasmosis, but not incur in self-citation as is the case [1, 2].
Line 86 correct Histoplasma ohiensi to Histoplasma ohiense
Line 125 correct H. ohiensi to H. ohiense
Materials and Methods
Please include the same changes of the previous sections.
One of my worries with the present work if the type of inoculum used, mycelial propagules are unreliable, because not all the mycelial fragments (depending of its size) the big ones are unable to reach the alveoli and produce infection. That is the reason that most researchers use mycelial microconidia, or yeast. Although in the material and methods it specifies that the inoculum is mainly microconidia, it should indicate what is the percentage, 60% - 90%?
In the numeral 2.2 Production of infective M-phase propagules. Please indicate the percentage of microconidia vs hyphal fragments.
Results
Apply the same changes naming the Histoplasma species used in the present work.
Discussion and Conclusions
As in the previous sections, Histoplasma spp. should be use and drop the use of Phylogenetic species, genotypes etc. Use H. suramericanum and H. ohiense and if prefer use additionally (EH-46, Lam A2) (G-217B, Nam 2), as a follow-up to the previous classifications.
The process of infection in Histoplasmosis is really complex, not only the host response is involved, but also the fungal specie, the size of the inoculum, as well as the phase of the dimorphic fungus are important. Although the mycelial phase is more natural, the massive dose of infection is non-natural, as indicated in references 3 and 16, which shows that using a high inoculum 5 x 105 H. ohiense (G-217B) is more virulent but at low doses H. capsulatum (G186A) have higher virulence. In the section from line 448 – 460, it is not clear that the virulence is related to the size of the inoculum.
Since the present study doesn’t evaluate the cfu in the murine model is difficult to define the virulence of the Histoplama species. Further studies may find virulence factors involve in the pathogenicity of the several defined species.
No comments
Author Response
Manuscript (ID: jof-2491620): Answers to the reviewers
Rev 1
Comments and Suggestions for Authors
Titles
The present article “Unusual Differences in the Pulmonary Histopathology of Mice after Intranasal Infection with Mycelial Propagules of Histoplasma capsulatum Phylogenetic Species from Latin America (LAm A2) and North America (NAm 2)”, had many problems beginning from the Titles.
1-Please change the title to:
Differences in the Pulmonary Histopathology of Mice after Intranasal Infection with Mycelial Propagules of Histoplasma spp. from Histoplasma suramericanum (LAm A2) and Histoplasma ohiense (NAm 2).
Answer: The authors do not agree to change the title by combining the names Histoplasma suramericanum (LAm A2) and Histoplasma ohiense (NAm 2) due to several scientific reasons and published data. The authors considered that this particular reviewer´s suggestion includes several misinterpretations, related to different understandings about the H. capsulatum phylogeny and taxonomy (Please, see the specific comments below).
Specific Comments:
The literature about Histoplasma phylogeny and current taxonomy is changing.
In fungal databases such as the Index Fungorum [www.indexfungorum.org] and the MycoBank [www.mycobank.org], the genus Histoplasma is classified only in two taxonomic species: H. capsulatum (which previously included the varieties H. capsulatum var. capsulatum and H. capsulatum var. farciminosum) and H. duboisii (previously known as H. capsulatum var. duboisii). Today the recognition of different varieties in the Histoplasma genus is considered obsolete [Please see: Mittal et al. 2019 (Curr Top Microbiol Immunol 422:157–191); Taylor et al. 2022 (Appl Environ Microbiol 88:e0201021; and also, the Index Fungorum and MycoBank databases].
Currently, one of the most important information about the Histoplasma genus is its phylogenetic classification published by Kasuga et al. 2003 (Mol Ecol 12:3383-33401). Originally, these authors have proposed that H. capsulatum should be considered a complex of cryptic species. Thus, several cryptic species are nested within the Histoplasma capsulatum complex. Kasuga et al. (2003) discriminated 8 clades (genetic populations), where seven of them are considered phylogenetic species (it should be noted that this classification is used at the present time to identify Histoplasma phylogenetic species with modifications implemented by Teixeira et al. 2016 (PLoS Negl Trop Dis 10:e0004732); Rodrigues et al. 2020 (Stud Mycol 97:100095); and Taylor et al. 2022 (Appl Environ Microbiol 88:e0201021). According to Taylor et al. (2022), H. capsulatum is a complex of different species containing at least 14 phylogenetic species and/or cryptic lineages globally distributed, where 11 of them are from the Americas. In the last 20 years, a lot of novel data about the H. capsulatum complex have been published and accepted as updated.
New information on the cryptic speciation of Histoplasma may lead to further refinements of Histoplasma classification, according to Sepúlveda et al. 2017 (MBio 5:8(6):e01339-17). Histoplasma taxonomy published by Sepúlveda et al. (2017) used a concatenated phylogenetic reconstruction in a whole genome assembly study and they renamed four Histoplasma geographical groups from the American continent, previously identified by Kasuga et al. (2003). As you know, Sepúlveda et al. (2017) classification proposed the following groups: H. capsulatum sensu stricto Darling 1906 (instead of lineage H81 from Panama); H. mississippiense sp. nov. (instead of NAm 1); H. ohiense sp. nov. (instead of NAm 2); and H. suramericanum sp. nov. (instead of LAm A). Although the methodology used by Sepúlveda et al. (2017) was very robust, it has some weaknesses, such as those explained by Voorhies et al. (2022) (mBio 13(1):e02574-21), who consider that there are some misinterpretations in the whole genome assembly studies due to genome fragmentation, which have led to underutilization of the genome-scale data. In addition, although all data published by Sepúlveda et al. (2017) are very interesting, in the case of the proposed taxonomic species H. suramericanum sp. nov., this name seems to us unsuitable, because it is not representative of the wide genetic diversity of the Histoplasma populations from South America and, particularly, from all Latin American regions (which included Histoplasma isolates from other countries that are geographically separated from South America). On the other hand, they only used strains from Colombia to propose the new species H. suramericanum sp. nov., which isn’t recommended for the most diversified Histoplasma population (LAm A clade) described in Latin America by Kasuga et al. 2003, and validated by Teixeira et al. 2016, Rodrigues et al. 2020, and Vite–Garin et al. J Fungi 7:529, 2021.
Thus, at this moment, we absolutely do not accept the nomenclature proposed by Sepúlveda et al. (2017) for the NAm 2 and LAm A phylogenetic species AS MANDATORY, based on the following precise explanations and experimental data: 1) Sepúlveda et al. classification with new Histoplasma nomenclatures are not considered as valid in the most important Fungal databases until now; 2) their proposal needs to be validated by several authors who work with Histoplasma taxonomy, using a critical number of isolates from the Latin American clades LAm A and LAm B (the latter with two monophyletic clades of Histoplasma, LAm B1 and LAm B2, according to Teixeira et al. 2016); 3) according to our strict laboratory rules, we only catalog taxonomically the Histoplasma isolates from our research group after submitting them to any type of genetic analysis, and as the EH-46 strain used in our paper was not phylogenomically analyzed, it is not recommendable to classify it as H. suramericanum; 4) to support the abovementioned idea, a recent paper published by Almeida-Silva et al. (J Fungi 7:865, 2021) has demonstrated a significant genetic diversity among several Histoplasma isolates that they classified as H. suramericanum, which confirms that this taxonomic species needs to be revised; 5) and, finally, to this date, there is no strong scientific data that demonstrates a mistake or a misinterpretation when a published study only refers to the phylogenetic species of the Histoplasma genus.
In conclusion, we ask for the reviewer to consider our point of view about the suggested modifications in the title and in the text of our manuscript that we are submitting to JoF, concerning the inclusion of Sepúlveda et al. (2017) nomenclature for the H. capsulatum strains used.
2-Another of my worries with the present work if the type of inoculum used, mycelial propagules are unreliable, because not all the mycelial fragments (depending of its size) are able to reach the alveoli and produce infection, that is the reason that most researchers use mycelial microconidia, or yeast. Although in the material and methods it specifies that the inoculum is mainly microconidia, it should indicate what is the percentage, 60% - 90%?
Answer: Regarding the type of inoculum used, the authors disagree with the reviewer´s comment for several reasons: 1) during a couple of years, we have been standardizing the use of M-phase infective propagules to reproduce “natural conditions of infection” in experimental histoplasmosis, trying to avoid any possible misinterpretations associated with the incorrect use of the infective fungal morphotype in the first step of the host immune response and with the use of an inappropriate via of infection; 2) it is also important to clarify that the size of the mycelial propagules isn´t a controversial problem because both microconidia and hyphal fragments present in our inoculum share similar sizes with the yeast cells that are used by other researches; 3) we agree with your comment that “not all the mycelial fragments (depending of its size) are able to reach the alveoli and produce infection”, however, with or without lungs involvement, the outcome of the infection after this first contact could depend on the inoculum size, the virulence of the Histoplasma strain, and the fate of the first interaction between the upper respiratory lymphoid tissue of the host and the fungal infective propagules, as we tried to show in the present paper in accordance with results published by Suárez-Álvarez et al. (2019) (Note: overall, we think that our infective model in mice reproduces the natural infection of Histoplasma more appropriately; 4) in laboratory conditions, to reproduce the natural infection it was necessary to anesthetize each mice by intramuscularly injecting a mixture of ketamine (100 mg/kg)/xylazine (10 mg/kg) to facilitate intranasal infection, dispensing 10 μL of the inoculum into each animal nostril at intervals of approximately 30 s, until completing 40 μL. In addition, in the Material and Methods section, we mentioned that the strains were cultured for 3-4 weeks at 28 °C in potato-dextrose-agar, which is a medium that facilitates fungal sporulation. Each M-phase was harvested in saline solution (SS) and centrifuged at 800 x g for 10 min at 4 °C, to remove most macroconidia and large aggregates of mycelium, and only those inocula containing approximately 90% microconidia and 10% hyphal fragments were used to infect mice. With this detailed explanation, we hope to have cleared up your doubts. Please, see the details described in the Material and Methods section.
3- Regarding the part of “Histoplasma capsulatum Phylogenetic Species” it doesn’t make any sense. Should be: “Histoplasma spp.” The two isolates used in the present work have been classified as H. ohiense (G-217B, Nam 2) and H. suramericanum (EH-46, Lam A2).
Answer: As it has been mentioned by several authors, Histoplasma capsulatum (NOT THE Histoplasma GENUS) has been considered a complex of cryptic species. We accept that under systematic concepts, in some way, this idea could become slightly complicated. However, in other microorganisms the same organization occurs, i.e., in Bacteria, as professors of Medical Bacteriology in the School of Medicine, at UNAM, we talk about the Mycobacterium tuberculosis complex (which included different Mycobacterium species, like M. tuberculosis, M. africanum, M. pinnipedii, M. caprae, M. bovis and others) and the Mycobacterium avium complex (which included M. avium, M. intracellulare). Thus, in Systematic studies, where cladistic or phylogenetic methods to classify microorganisms is used, it is not erroneous to say “Histoplasma capsulatum Phylogenetic Species”. In order to avoid conflicting views, we decided to modify the title of the paper to “Unusual Differences in the Pulmonary Histopathology of Mice after Intranasal Infection with Mycelial Propagules of Histoplasma capsulatum Strains Classified as LAm A2 and NAm 2 Phylogenetic Species”.
4- The reference 11(Sepulveda VE, Marquez R, Turissini DA, Goldman WE, Matute DR. Genome Sequences Reveal Cryptic Speciation in the Human Pathogen Histoplasma capsulatum. mBio. 2017;8(6). 10.1128/mBio.01339-17), that use complete genomes of Histoplasma, validates the original classification of Kasuga et al. (Ref. 9), but give proper names to several of the original clades described by Kasuga et. al. and should be use, at least the authors of the present work have data supporting the actual classification presented in reference 11.
Answer: Because of the reasons we have provided in response to your first question, these modifications were not included in the manuscript text.
Abstract
5- The same changes should be done in the abstract, using Histoplasma spp. instead of Histoplasma capsulatum. Also use H. suramericanum (EH-46, Lam A2) and H. ohiense (G-217B, Nam 2) and drop the use of Phylogenetic species since these species have already been described.
Answer: Because of the reasons we have provided in response to your first question, these modifications were not included in the manuscript text. In addition, in a previous paper published by Vite-Garin et al. (J Fungi 7:529, 2021), you can see that our point of view is shared by other colleagues.
6- At the end of the abstract change Histoplasma genotypes for Histoplasma species.
Answer: OK. See the correction done in the revised version of the manuscript.
Introduction
7- Similar changes should be introduced in this section, referring to Histoplasma spp. or Histoplasma strains. Drop the use of Phylogenetic species that don’t make any sense, as mentioned above.
Answer: Because of the reasons we have provided in response to your first question, these modifications were not taken into consideration.
8- In the initial part, where the description of the disease and its clinical manifestations are made, the reference that should be used, is one related with the subject, such a recent Medical Mycology book, or good book on infectious diseases, in its defect a review on Histoplasmosis, but not incur in self-citation as is the case [1, 2].
Answer: We are very sorry, but you are incorrect. We cited three articles [references 1-3], including one by Sepúlveda et al. (2014), supporting several clinical manifestations in histoplasmosis because they briefly summarize this matter. Honestly, we prefer not to question or make an issue neither out of this comment nor of your recommendation to use the Medical Mycology book as a source of reference.
9- Line 86 correct Histoplasma ohiensi to Histoplasma ohiense.
Answer: OK. See the corrections in the revised version of the manuscript.
10- Line 125 correct H. ohiensi to H. ohiense.
Answer: OK. See the corrections in the revised version of the manuscript.
Materials and Methods
11- Please include the same changes of the previous sections.
Answer: Because of the reasons we have given in response to your first question, there were no Histoplasma taxonomic changes included in the manuscript text.
12- One of my worries with the present work if the type of inoculum used, mycelial propagules are unreliable, because not all the mycelial fragments (depending of its size) the big ones are unable to reach the alveoli and produce infection. That is the reason that most researchers use mycelial microconidia, or yeast. Although in the material and methods it specifies that the inoculum is mainly microconidia, it should indicate what is the percentage, 60% - 90%?
Answer: This question is duplicated. Please, see the details described in the Material and Methods section and in our response to your second question.
13- In the numeral 2.2 Production of infective M-phase propagules. Please indicate the percentage of microconidia vs hyphal fragments.
Answer: Please, see the details described in the Material and Methods section and in our response to your second question.
Results
14- Apply the same changes naming the Histoplasma species used in the present work.
Answer: In accordance with our reasons described in response to your question number 1, these modifications in the manuscript text were not adopted. Because of the reasons we have given in response to your first question, these modifications were not included in the manuscript text.
Discussion and Conclusions
15- As in the previous sections, Histoplasma spp. should be use and drop the use of Phylogenetic species, genotypes etc. Use H. suramericanum and H. ohiense and if prefer use additionally (EH-46, Lam A2) (G-217B, Nam 2), as a follow-up to the previous classifications.
Answer: Because of the reasons we have given in response to your first question, these modifications were not taken into consideration.
16- The process of infection in Histoplasmosis is really complex, not only the host response is involved, but also the fungal specie, the size of the inoculum, as well as the phase of the dimorphic fungus are important. Although the mycelial phase is more natural, the massive dose of infection is non-natural, as indicated in references 3 and 16, which shows that using a high inoculum 5 x 105 H. ohiense (G-217B) is more virulent but at low doses H. capsulatum (G186A) have higher virulence. In the section from line 448 – 460, it is not clear that the virulence is related to the size of the inoculum.
Answer: We would like to remind you that about the references 3 and 16, which are from other authors, the interpretation of their results was done by them. Thus, we prefer not to modify this in the Discussion text. However, it is important to highlight that the authors of reference 3 (Sepúlveda et al. 2014) and reference 16 (Jones et al. 2020) have identified differences in fungal virulence, irrespective of the inoculum size of the H. capsulatum strains tested. In fact, in other reports by different authors, the H. capsulatum G-186A or G-186B strains always showed higher virulence than the G-217B strain.
17- Since the present study doesn’t evaluate the cfu in the murine model is difficult to define the virulence of the Histoplama species. Further studies may find virulence factors involve in the pathogenicity of the several defined species.
Answer: We believe that you have misinterpreted what we are saying about the host (mice) inflammatory response in regard to H. capsulatum infection with two genetically distinct strains. We never talk about differences in virulence between the two distinct H. capsulatum strains used in the paper that we are submitting. We only mentioned that “our findings highlight differences in the host pulmonary inflammatory response associated with distinct H. capsulatum species” and that our “results may contribute new information to better understand H. capsulatum-induced lung inflammatory reactions associated with pneumonia, granuloma, or BALT induction” (Please, revise the Abstract, Results, Discussion, and Conclusion sections of our paper).
Comments on the Quality of English Language
No comments
Reviewer 2 Report
Overall, you have done an impressive study and presented results to bring out aspects of early-to-late phenomena in infection with two different strains of Histoplasma, and your data highlight the differences in host responses to the two strains. I would have liked to see some biochemical or immunological correlates of the early-to-late events in parallel with the visual representation via histopathology, but absent that, this paper stands on its own to showcase the tissue responses.
The current manuscript needs a few edits and updates. For instance:
Line 48-49: "environments containing bat and bird guano that favor M-phase growth" -- needs reference(s).
Line 53; "nevertheless" does not make sense since granuloma formation (described in the previous sentence) is a function of host's immune status.
Line 68: Change to "A few" which has a different meaning than "Few" in English.
line 69: The first part of the sentence is unclear in meaning. I recommend you state the exact finding by Suarez-Alvarez.
Line 110: "should not be questionable" is an odd phrase to write in a scientific paper. I recommend you rephrase.
Line 159: Question: How were microconidia and hyphal fragments counted in a hemocytometer and how did you ascertain that the numerical value assigned by you to the count was correct?
Line 192. Question: Did you ever consider using another stain along with PAS, e.g., GMS? If not, why not?
For all figures: in the description and legends it is not clear whether the panels shown are representative of the mice you tested at each time point. What about the relative distributions of the reported features for the two strains that were studied? (What I mean is, for each feature you have reported, was there more or less of the feature observed in EH-46 vs. G217B?
For all figures: I suggest you follow a uniform scheme for all the panels. Show both EH-46 and G217B (not just EH-46), putting EH-46 on the left and G217B on the right. Wherever you have shown blow-up photos, place them below the respective panel (and not to the right side). Otherwise, the images are getting too confusing to comprehend what is being shown.
Please indicate the magnifications for all panels in the figure legends.
Please ensure that the figures represent the best features from all the mice you have tested at each time point for each Histoplasma strain.
Figure 1 needs more panels.
Figure 2 needs better photos at different magnifications.
Line 368: provide some sort of enumeration to demonstrate frequency of occurrence of a feature.
Discussion: refocus the generalized discussion on granulomas to specific observations with the two strains.
Line 396. What does "registered the presence" mean? Define method in Methods and present results in Results.
Line 407: rephrase "PAS-positive structures like macroconidia" because macroconidia are not the only PAS-positive structures for Histoplasma, and you have indeed seen PAS+ structures beyond 3 hours (e.g., yeast cells).
This iteration of the manuscript requires extensive editing for English language. There are occasional odd choices of words, phrases, or expressions, as well as grammatical errors, sprinkled throughout, which are unfortunately making the narrative difficult to understand and detracting from the value of this paper. I recommend that the authors seek assistance from a knowledgeable native speaker of English or some English language service.
Author Response
Manuscript (ID: jof-2491620): Answers to the reviewers
Rev 2
Comments and Suggestions for Authors
Overall, you have done an impressive study and presented results to bring out aspects of early-to-late phenomena in infection with two different strains of Histoplasma, and your data highlight the differences in host responses to the two strains. I would have liked to see some biochemical or immunological correlates of the early-to-late events in parallel with the visual representation via histopathology, but absent that, this paper stands on its own to showcase the tissue responses.
The current manuscript needs a few edits and updates. For instance:
1-Line 48-49: "environments containing bat and bird guano that favor M-phase growth" -- needs reference(s).
Answer: Please, take into account that the above-mentioned phrase reproduces classic knowledge in histoplasmosis that is routinely documented; hence, the decision of the authors not to include any reference regarding this matter was because we did not deem it necessary. To clear up any doubts the reviewer may have, we next mention two recently published papers, where bat and bird guano is mentioned in relation to risk factor involved in histoplasmosis: 1) Gómez LF; Arango M; McEwen JG; Gómez OM; Zuluaga A; Peláez CA; Acevedo JM; Taylor ML; Jiménez MP: Molecular epidemiology of Colombian Histoplasma capsulatum isolates obtained from human and chicken manure samples. Heliyon 5:e02084, 07/2019, doi: 10.1016/j.heliyon.2019.e02084; 2) Gómez-Londoño LF; Pérez-León LC; McEwen Ochoa JG; Zuluaga Rodriguez A; Peláez-Jaramillo CA; Acevedo-Ruiz JM; Taylor ML; Arango-Arteaga M; Jiménez-Alzate MP: Capacity of Histoplasma capsulatum to survive the composting process. Appl. Environ. Soil Sci. 2019:5038153, 09/2019, doi: 10.1155/2019/5038153.
2-Line 53; "nevertheless" does not make sense since granuloma formation (described in the previous sentence) is a function of host's immune status.
Answer: OK. Please, see the correction in the revised version.
3-Line 68: Change to "A few" which has a different meaning than "Few" in English.
Answer: OK. Please, see the correction in the revised version.
4-line 69: The first part of the sentence is unclear in meaning. I recommend you state the exact finding by Suarez-Alvarez.
Answer: Please, see the correction in the first part of the sentence. From our point of view, the correction exactly describes the importance of the findings reported by Suárez-Álvarez et al. (2019).
5-Line 110: "should not be questionable" is an odd phrase to write in a scientific paper. I recommend you rephrase.
Answer: In attention to your comment we have rewritten this sentence. Please, see the following correction “and the genetic attributes of the fungus involved in the host´s infection should be considered as a relevant factor in the infective process.
6-Line 159: Question: How were microconidia and hyphal fragments counted in a hemocytometer and how did you ascertain that the numerical value assigned by you to the count was correct?
Answer: In the Material and Methods section, we mentioned that the strains were cultured for 3-4 weeks at 28 °C in potato-dextrose-agar, which is a medium that facilitates fungal sporulation. Each M-phase was harvested in saline solution (SS) and centrifuged at 800 x g for10 min, at 4 °C, to remove most macroconidia and large aggregates of mycelium, and only those inocula containing approximately 90% microconidia and 10% hyphal fragments were used to infect mice. Each inoculum was counted in a hemocytometer by optical microscopy and adjusted to 3 × 106 infective propagules/40 µL. Inoculum viability was determined by the trypan blue (0.5%) exclusion test and the infective inoculum was confirmed by culturing on non-supplemented BHI-agar (Bioxón) at 28◦C, using colony forming unit determinations. With this detailed explanation, we hope to have clarified your doubts (Please, see the Material and Methods section of the revised version of the manuscript).
7-Line 192. Question: Did you ever consider using another stain along with PAS, e.g., GMS? If not, why not?
Answer: Periodic acid Schiff (PAS) and Grocott (or Gomori) Methenamine Silver (GMS) stains clearly show fungi in the histological section. Their usefulness as ancillary methods for staining fungal infections is widely accepted. Less well-studied is the comparison in sensitivity and specificity between these two stains. Numerous studies support the frequent use of PAS in the detection of fungal infections. In addition, although some studies suggest that GMS may outperform PAS in detecting fungus, GMS is generally considered more expensive and it requires more technical precision than PAS. We chose to use PAS in the present work because, in our experience, it allows us to identify a great variety of histopathologic inflammatory reaction patterns, including the granuloma formation, in a better way than GMS stain, which can often result in a high contrast (black against a green background, with connective tissue elements also appearing black stained) that prevents histopathological analyses as well as fungus identification.
8-For all figures: in the description and legends it is not clear whether the panels shown are representative of the mice you tested at each time point. What about the relative distributions of the reported features for the two strains that were studied? (What I mean is, for each feature you have reported, was there more or less of the feature observed in EH-46 vs. G217B?
Answer: Please, see the correction in the revised version. The histopathologic findings showed a diffuse distribution in lung sections, but mice infected with the EH-46 strain developed more focal inflammatory responses (granulomas) than G-217B-infected mice, mainly after 7 days postinfection.
9-For all figures: I suggest you follow a uniform scheme for all the panels. Show both EH-46 and G217B (not just EH-46), putting EH-46 on the left and G217B on the right. Wherever you have shown blow-up photos, place them below the respective panel (and not to the right side). Otherwise, the images are getting too confusing to comprehend what is being shown.
Answer: We are so sorry; we have different view about the order in which we are presenting the figures. All figures have EH-46 and G-217B photomicrographs. Most of them follow the sequence that you suggested (“putting EH-46 on the left and G217B on the right”), except figures 1 and 6. Figure 1 corresponds to findings observed 1 h postinfection and very few histopathological changes and fungal structures were found. Figure 6 corresponds to findings observed 14 days postinfection, where mice infected with the EH-46 strain showed prominent granuloma developments and, in contrast, mice infected with the G-217B strain mainly showed BALTs. For these reasons, we organized the panels according to the relevance of our findings, as you can see in the manuscript.
10-Please indicate the magnifications for all panels in the figure legends.
Answer: Please, see the scale bars showing the magnifications in the figures. In addition, each magnification of the scale bars is showing in each figure legend (see the revised version of the manuscript).
11-Please ensure that the figures represent the best features from all the mice you have tested at: each time point for each Histoplasma strain.
Answer: Different histopathological events were carefully analyzed in the figures and only those deemed as more representative were chosen by the authors of this manuscript. Figures 6 and 7 were modified to make the histopathological findings clearer.
12-Figure 1 needs more panels.
Answer: Please, see the answer to question number 9.
13-Figure 2 needs better photos at different magnifications.
Answer: From our point of view, we do not believe it is necessary to include more photomicrographs with different magnifications in figure 2. Could you, please, explain to us why you think so?
14-Line 368: provide some sort of enumeration to demonstrate frequency of occurrence of a feature.
Answer: The present study follows a descriptive approach, thus in the manuscript text, frequencies are referred to make qualitative comparisons between the histopathological findings obtained with the two Histoplasma strains used.
15-Discussion: refocus the generalized discussion on granulomas to specific observations with the two strains.
Answer: Please, see the rewritten text in the Discussion section, regarding the role of granuloma formation in histoplasmosis. We hope to have succeeded refocusing our idea about granuloma involvement in histoplasmosis. Also, we would like to reinforce our idea through the following considerations: According to our histopathological observations, after 14 and 21 days postinfection, granuloma development was clearly associated with the EH-46 H. capsulatum strain infection, whereas BALT induction, with a lesser diffuse inflammatory response, was associated with the G-217B strain infection. These findings support the existence of relevant differences in the pulmonary histopathology induced by each of the two H. capsulatum strains used in the present study, which could be crucial in the formation of granulomas for mice infected with the EH-46 strain. In addition, at least in our study, the development of pulmonary granulomas does not seem to depend on the genetic characteristics of the host. Thus, our results could encourage the undertaking of further studies to investigate the virulence of Histoplasma strains in the clinical evolution of an experimental infection. In addition, obviously, in the future it would be interesting to describe the role of fungal virulence factors in clinical manifestations of histoplasmosis and, particularly, in the activation pathways of the host cells, which could be engaged in the host immune response for the formation of pulmonary granulomas in an experimental model with H. capsulatum.
16-Line 396. What does "registered the presence" mean? Define method in Methods and present results in Results.
Answer: Please, see the corrections done in different parts of the manuscript text, where the word “registered” was replaced by a more adequate word.
17-Line 407: rephrase "PAS-positive structures like macroconidia" because macroconidia are not the only PAS-positive structures for Histoplasma, and you have indeed seen PAS+ structures beyond 3 hours (e.g., yeast cells).
Answer: We are very sorry; we find your observation very confusing.
Comments on the Quality of English Language
This iteration of the manuscript requires extensive editing for English language. There are occasional odd choices of words, phrases, or expressions, as well as grammatical errors, sprinkled throughout, which are unfortunately making the narrative difficult to understand and detracting from the value of this paper. I recommend that the authors seek assistance from a knowledgeable native speaker of English or some English language service.
Answer: The first version of the manuscript that we sent to JoF was revised by “The American Journal Experts (AJE)” as can be attested by the certificate they issued, which may be verified on the AJE website, using the verification code 5114-C494-595A-4364-417D. However, in view of your recommendation, the revised version of the manuscript that we are submitting to JoF will once more be sent to a proofreader of English texts.
Round 2
Reviewer 1 Report
I still have some worries about the paper, but a manuscript doesn't have to be perfect to be published.